# Designing microbial communities to maximize the thermodynamic driving force for the production of chemicals

**Pavlos Stephanos Bekiaris**, **Steffen Klamt***

Max Planck Institute for Dynamics of Complex Technical Systems, Magdeburg, Germany

* klamt@mpi-magdeburg.mpg.de

## Abstract

Microbial communities have become a major research focus due to their importance for bio-geochemical cycles, biomedicine and biotechnological applications. While some biotechnological applications, such as anaerobic digestion, make use of naturally arising microbial communities, the rational design of microbial consortia for bio-based production processes has recently gained much interest. One class of synthetic microbial consortia is based on specifically designed strains of one species. A common design principle for these consortia is based on division of labor, where the entire production pathway is divided between the different strains to reduce the metabolic burden caused by product synthesis. We first show that classical division of labor does not automatically reduce the metabolic burden when metabolic flux per biomass is analyzed. We then present ASTHERISC (Algorithmic Search of THERmodynamic advantages in Single-species Communities), a new computational approach for designing multi-strain communities of a single-species with the aim to divide a production pathway between different strains such that the thermodynamic driving force for product synthesis is maximized. ASTHERISC exploits the fact that compartmentalization of segments of a product pathway in different strains can circumvent thermodynamic bottlenecks arising when operation of one reaction requires a metabolite with high and operation of another reaction the same metabolite with low concentration. We implemented the ASTHERISC algorithm in a dedicated program package and applied it on *E. coli* core and genome-scale models with different settings, for example, regarding number of strains or demanded product yield. These calculations showed that, for each scenario, many target metabolites (products) exist where a multi-strain community can provide a thermodynamic advantage compared to a single strain solution. In some cases, a production with sufficiently high yield is thermodynamically only feasible with a community. In summary, the developed ASTHERISC approach provides a promising new principle for designing microbial communities for the bio-based production of chemicals.

**Data Availability Statement:** Data are available within the manuscript and Supporting Information and in parts from a public repository: https://github.com/klamt-lab/CommModelPy https://github.com/klamt-lab/astheriscPackage.

**Funding:** This work was supported by the EU program ERDF (European Regional Development Fund) of the German Federal State of Saxony-Anhalt within the Research Center for Dynamic Systems (CDS; given to SK), and by the German Federal Ministry of Education and Research (FKZ: 031B0524B; given to SK). The funders had no role in study design, data collection and analysis, decision to publish, or preparation of the manuscript.

**Competing interests:** The authors have declared that no competing interests exist.

## Author summary

Communities of microbes are ubiquitous in nature and also of high relevance for industrial applications, e.g. for the production of biogas. The development and use of non-natural communities for biotechnological applications has become an important subject of research. In this work, we present a new computational method to design synthetic communities with improved capabilities for the synthesis of desired target metabolites. Our method takes a constraint-based metabolic model of an organism as input and searches for a suitable partitioning of the product pathway via different strains of the organism such that the thermodynamic driving force for product synthesis is maximized. Essentially, this approach exploits the fact that having multiple strains allows adjustment of different metabolite concentrations in the different strains by which the thermodynamic driving force for product synthesis can often be increased. We tested this approach with a core and with a genome-scale metabolic network model of *Escherichia coli*. We found that, for dozens of metabolites, there exist communities with specifically designed strains of *E. coli* where the maximal thermodynamic driving force can be increased compared to a single *E. coli* strain. In summary, our presented method provides a new approach, together with a new design principle, for the computational design of microbial communities.

## Introduction

In nature, organisms rarely occur as isolated populations of single species. They rather form communities (or consortia) with different types of complex interactions between the participating species which often drive evolution [1]. In particular, communities of microorganisms are ubiquitous in nature and play a fundamental role for ecology, bioremediation, geochemical cycles and human health. Microbial communities typically consist of different species, however, especially under controlled laboratory conditions, they may also comprise coexisting strains of a single species with certain physiological differences between the strains. One way to establish such single-species (multi-strain) communities is to construct and co-culture different, genetically modified strains with obligate mutual metabolic dependencies [2].

Mathematical modeling has become a valuable tool to formally describe and simulate complex microbial communities and to gain a deeper understanding of their behavior and properties. A range of different methods has been developed for community modeling [3–5]. As for modeling single species, two major approaches are kinetic modeling based on differential equations [6,7] and steady-state modeling utilizing constraint-based (flux balance analysis (FBA) [8]) techniques [9–13]. FBA-related methods are limited to predictions on stationary metabolic fluxes, but they require as input only the metabolic reaction network of the participating species and no data on kinetic mechanisms and parameters and can therefore be used to analyze large (genome-scale) metabolic models. Applying (linear) FBA-techniques to study communities models requires partially adaptations, e.g. due to arising bilinearities. Two particular examples of such formalizations are SteadyCom [14] and RedCom [15]. Both methods are based on the concept of balanced growth of a community [16] and allow predictions of growth rates, metabolic exchange rates and feasible community compositions. An example of an FBA-based method for the design of single-species (multi-strain) communities with certain constraints is DOLMN (Division Of Labor In Metabolic Networks) [17].

Microbial consortia are extensively used in biotechnological applications. One example is anaerobic digestion in biogas plants where methane is produced by naturally occurring

communities. In recent years, artificial co-cultures have been constructed for synthesizing bio-fuels and value-added products in bioreactors [18]. While some of these realizations use multi-species communities [19,20], more and more realizations focus on the use of communities with different, purposefully constructed strains of a single species. Successful examples of single-species communities for product synthesis include (a) a two-strain *E. coli* community for the conversion of xylan to ethanol with high ethanol yields [21], (b) another two-strain *E. coli* community for the synthesis of flavonoids [22] with much higher synthesis rates than without communities, and (c) the synthesis of anthocyanins with a community of four *E. coli* strains [23], the first successful production of this chemical by microbes at all.

The high potential of dedicated single-species communities for optimizing biotechnological production of a certain compound has been intensively discussed in recent literature. Division of labor (DoL) is seen as one major design principle of such artificial co-cultures [24–28]. The basic idea of DoL is illustrated with the toy example shown in Fig 1A, where we consider a linear metabolic pathway converting a substrate S into a desired target product P. Concretely, S is taken up, then, in a first reaction catalyzed by enzyme $E_1$, converted to the intermediate metabolite M, which is in turn further converted to the product P in a second reaction catalyzed by enzyme $E_2$, after which P is finally excreted. In a biotechnological application, the goal is to achieve a high flux through this pathway which will require a high abundance of the enzymes $E_1$ and $E_2$. Due to limited intracellular resources, the synthesis of each of the two enzymes competes with each other and with other cellular processes, which may result in a reduced production performance of the strain. Here, DoL may help to reduce the "metabolic burden" of

**A) Basic idea of division of labor (DoL)**

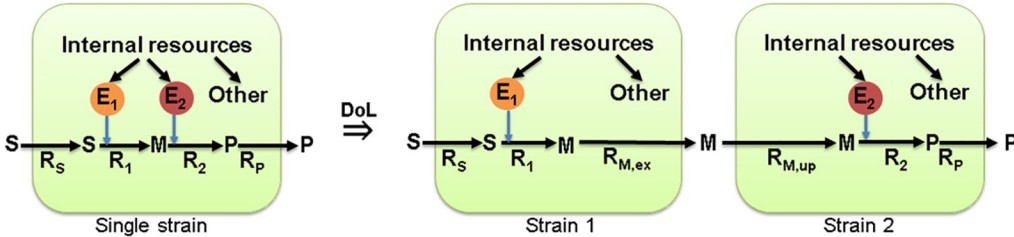

**B) DoL under consideration of biomass usage**

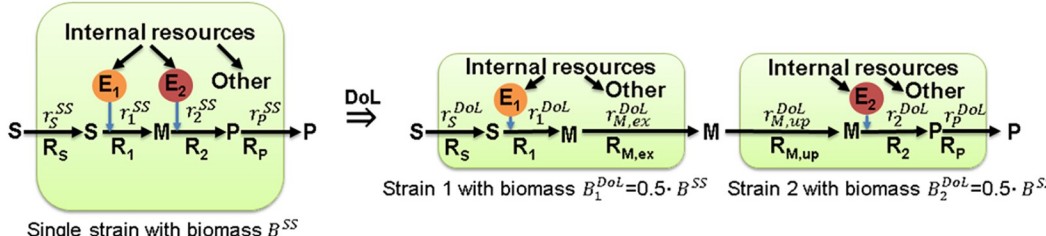

$$J_P^{SS} = B^{SS} \cdot r_P^{SS} = B^{SS} \cdot r_1^{SS} = B^{SS} \cdot r_2^{SS} \qquad J_P^{DoL} = B_1^{DoL} \cdot r_1^{DoL} = 0.5 \cdot B_1^{SS} \cdot r_1^{DoL} = B_2^{DoL} \cdot r_2^{DoL} = 0.5 \cdot B_2^{SS} \cdot r_2^{DoL}$$

$$J_P^{SS} = J_P^{DoL} \Rightarrow [E_1]^{DoL} = 2 \cdot [E_1]^{SS}, \ [E_2]^{DoL} = 2 \cdot [E_2]^{SS},$$
$$[E_{tot}]^{SS} = [E_1]^{SS} + [E_2]^{SS} = [E_1]^{DoL} = [E_2]^{DoL}$$

**Fig 1. An example illustrating that DoL does not reduce the overall metabolic burden.** (A) Basic idea of DoL: a two-step pathway converting a substrate S to a product P via the two reactions $R_1$ (catalyzed by enzyme $E_1$) and R2 (catalyzed by enzyme $E_2$) in a single strain is split into two parts each being performed by one dedicated strain. (B) Assuming an identical total amount of biomass, the metabolic burden (enzyme cost per biomass) is identical for the single strain and the DoL solution.

the single strain by dividing the respective metabolic task and thus the metabolic costs on several strains of the same species [24,29]. In our example, a DoL strategy could be to split the pathway into two parts (or tasks) each being implemented in one dedicated strain. With an exchange of the intermediate metabolite M, each of the two strains may now concentrate on its dedicated task which could circumvent physiological bottlenecks. Clearly, the effectiveness of a DoL strategy will depend on the relative burden on each pathway step. However, even in the ideal case that the metabolic burden is equally distributed over the two reactions and thus over the two different strains, this DoL perspective neglects one major constraint: if we assume that we are given a certain amount of biomass that can be used for product synthesis (higher amounts of biomass will inevitably reduce product yield), DoL will not necessarily lead to increased product flux (Fig 1B). In the single strain approach, the specific rates (in [mmol/(gDW·h)]) for reaction $R_1$ are given by

$$r_1^{SS} = [\text{E}_1]^{SS} \cdot k_{cat,1} \cdot f_1(\boldsymbol{c}) \tag{1}$$

where $[\text{E}_1]^{SS}$ is the concentration of enzyme $E_1$ in the single strain, $f_1$ the saturation function characterizing the kinetic rate law of reaction $R_1$, and $\boldsymbol{c}$ the vector of the metabolite concentrations. Analogously, for the specific rate of reaction $R_2$ we have:

$$r_2^{SS} = [\text{E}_2]^{SS} \cdot k_{cat,2} \cdot f_2(\boldsymbol{c}) \tag{2}$$

For simplicity, in the following we assume that the same amount of enzyme $E_1$ and $E_2$ is needed to induce a certain flux in the reactions $R_1$ and $R_2$ (implying $k_{cat,1} \cdot f_1(\boldsymbol{c}) = k_{cat,2} \cdot f_s(\boldsymbol{c})$), and that both enzyme costs are identical (S1 Text shows that the same conclusions can be drawn in the more general case with arbitrary pathway length and enzyme costs under assumption of constant metabolite concentrations in the strains). Therefore, since the specific rates in the single-strain solution must fulfill $r_S^{SS} = r_1^{SS} = r_2^{SS} = r_P^{SS}$ in steady state, the enzyme concentrations must be identical: $[\text{E}_1]^{SS} = [\text{E}_2]^{SS}$. For the total product synthesis flux $J_P^{SS}$ [mmol/h] we need to multiply the specific flux with the amount of biomass $B^{SS}$ (unit [gDW]): $J_P^{SS} = B^{SS} \cdot r_1^{SS} (= B^{SS} \cdot r_2^{SS})$. Now, in the DoL approach with the assumed simplified case of identical enzyme requirements and costs, the biomass of the single strain would be divided into two equal parts (strains 1 and 2, each having 50% of the original total biomass: $B_1^{DoL} = B_2^{DoL} = 0.5 \cdot B^{SS}$). For the steady-state product synthesis flux in the DoL approach, we obtain $J_P^{DoL} = B_1^{DoL} \cdot r_1^{DoL} = B_2^{DoL} \cdot r_2^{DoL}$. Therefore, in order to obtain the same total product synthesis flux as for the single-strain solution, the specific rates $r_1^{DoL}$ and $r_2^{DoL}$ must be doubled due to the halved biomass available for each strain. This requires doubled enzyme concentrations in the two strains, $[\text{E}_1]^{DoL} = 2[\text{E}_1]^{SS} = [\text{E}_2]^{DoL} = 2[\text{E}_2]^{SS}$, implying that the total concentration of enzymes required for the product pathway is identical in all three strains considered: $[\text{E}]_{tot}^{SS} = [\text{E}_1]^{SS} + [\text{E}_2]^{SS} = [\text{E}_1]^{DoL} = [\text{E}_2]^{DoL}$. Thus, the total enzyme costs per biomass in the DoL scenario are the same as with the single-strain solution and the metabolic burden is not reduced. Furthermore, the exchange of the intermediate metabolite M between the strains under DoL is also associated with additional metabolic costs (transporters must be produced, energetic costs of transport etc.). In fact, as also shown for the more general case (S1 Text), a DoL strategy cannot be advantageous (or is even unfavorable) with respect to metabolic burden as long as a potential kinetic advantage due to different metabolite concentrations in the different strains does not outweigh the added costs of metabolite transport. This point has rarely been considered when discussing DoL strategies.

As indicated in the last statement, one degree of freedom in a community that could be used to truly enhance the overall production rate, is the possibility of having different metabolite concentrations in the strains. In this study we present a new approach for designing and

**Fig 2. Example illustrating how division of labor may lead to a thermodynamic advantage in the production of a target metabolite.** In the left, a metabolic pathway in a cell is considered that synthesizes the target product P. The red values indicate positive values for the standard Gibbs free energy change ($\Delta_r G'^0$ [in kJ/mol]) and thus potential thermodynamic bottlenecks. With an allowed concentration range from 1 M to 10 M for all metabolites except for $P_{ex}$, where a minimum concentration of 5 M was assumed to consider product synthesis under high external product concentrations, a negative optimal MDF (OptMDF) value would follow, indicating thermodynamic infeasibility of product synthesis in the single strain. In the two-strain community (right), the pathway is divided and an exchange of metabolite B introduced. With this, individual concentrations of metabolite X can be adjusted in the two strains by which thermodynamic feasibility (a positive OptMDF) of the overall transformation is achieved (the blue triangles indicate the direction of the concentrations of X (high/ low) when maximizing the driving force). Black arrows in the two-strain solution indicate active and grey arrows inactive reactions.

optimizing communities with multiple strains of a single species, which exploits this degree of freedom. Specifically, our concept seeks to identify a suitable division of the metabolic network, such that the thermodynamic driving force of the entire product synthesis pathway is maximized. The basic concept is shown with the toy model in Fig 2: The pathway from substrate S to product P includes a thermodynamic bottleneck at the reactions $R_2$ and $R_5$ because of their relatively low thermodynamic driving forces (positive standard Gibbs free energies). The driving force of reaction $R_2$, where X is a reactant, could be increased by adjusting a lower concentration of metabolite X, however, this would simultaneously further reduce the driving force of reaction $R_5$, where X is a product. In fact, with the chosen set of constraints for the metabolite concentrations in Fig 2, the pathway is thermodynamically infeasible indicated by the negative optimal MDF (OptMDF) value. For a given pathway, the MDF (max-min-driving force [30]) maximizes the minimum of the driving force (i.e., the negative value of the Gibbs free energy change) of all participating reactions in a pathway and OptMDF finds the pathway in a metabolic network with maximal MDF [31]. Here, splitting the pathway in two subpathways and implementing them in two different strains with an exchange of intermediate B

(Fig 2) circumvents these bottlenecks as the concentration of metabolite X can now be adjusted individually in each strain (low in strain 1, high in strain 2) through which product synthesis becomes thermodynamically feasible. Following this idea, we developed a new framework, ASTHERISC, for an automated design of single-species (multi-strain) communities that enable higher thermodynamic driving forces for product synthesis compared to a single-strain approach. Focusing on pure production (without growth), we demonstrate the power of the method by identifying multi-strain communities of *E. coli* that maximize the thermodynamic driving force of product synthesis for a wide range of compounds.

## Methods

### Structure of community models

We first summarize the representation of stoichiometric models of microbial communities under balanced growth (for a detailed description can be found in [15]). Herein, we will focus on communities with different strains of one species, which can be seen as a special case of communities with multiple species. As common for constraint-based metabolic models we represent the metabolic network (with $m$ metabolites and $q$ reactions) of the given species with a $m \times q$ stoichiometric matrix $\mathbf{N}$, a flux vector $\mathbf{r}$ containing the net reaction rates, and upper ($\beta_i$) and lower ($\alpha_i$) flux bounds for each reaction. The steady-state assumption for the internal metabolites implies

$$\mathbf{Nr} = \mathbf{0} \tag{3}$$

and the flux bounds restrict the range of possible reaction rates:

$$\alpha_j \leq r_j \leq \beta_j \tag{4}$$

While the stoichiometric matrix $\mathbf{N}$ comprises internal metabolites only, this matrix can be extended to $\tilde{\mathbf{N}}$, which also contains the stoichiometries of the external metabolites (substrates, products etc.) which need not fulfill the steady state constraint (3). We also allow the definition of additional linear flux constraints (such as demanded minimal product-to-substrate yields) with a suitable matrix $\mathbf{D}$ and vector $\mathbf{d}$:

$$\mathbf{Dr} \leq \mathbf{d} \tag{5}$$

In the description of microbial communities, the net reaction rates $\mathbf{r}$ are normalized with respect to the total community biomass [mmol/(gDW$_{\text{total}}$· h)]. Furthermore, the biomass fractions $F_i = BM_i/BM_{tot}$ ($0 \leq F_i \leq 1$) of each strain $i$ at the total biomass are variables of the model. To fulfill the constraint of balanced growth (i.e., constant composition and identical specific growth rate for each strain), the ratio of biomass production of strain $i$ to total biomass synthesis must also equal $F_i$. Importantly, as the original flux bounds (4) for each strain are given in mmol/(gDW$_i$·h), they must be multiplied with the fraction of the respective strain in the community to obtain flux bounds with units normalized to total biomass

$$F_i \alpha_{ji} \leq r_{ji} \leq F_i \beta_{ji} \tag{6}$$

In communities with (balanced) growth, the formulation leads to bilinearities [15], which can be resolved by fixing either the strain fractions or the community growth rate in order to obtain a linear optimization problem. However, in this study we focus on optimal partitioning of the product synthesis pathway (without growth) and the only flux bound used will be the (community) uptake rate of the carbon source. With that we can disregard the production of biomass and assume identical fractions for all involved strain (e.g. 50% in two-strain

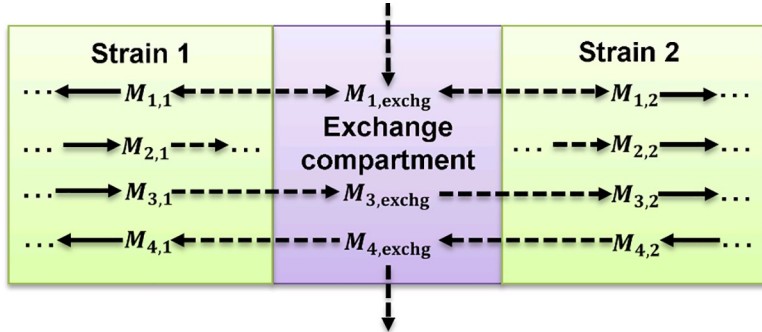

**Fig 3. Community model structure used in this study and possible exchange directions of metabolites.** Dashed arrows indicate exchange reactions, all other arrows biochemical conversions.

communities and 33% in three-strain communities; see S2 Text and Results section). However, in general, growth or/and different biomass fractions may be considered if this is desired.

In the community model, each of the $n$ strains represent a compartment with an associated metabolic network, captured by stoichiometric matrices $\mathbf{N_1}$ to $\mathbf{N_n}$. In a multi-strain model of one species, the $n$ stoichiometric matrices are, at least initially, identical. In addition, an exchange compartment is added from/to which the strains can uptake/release compounds (Fig 3). In particular, this enables the exchange of metabolites between strains. The quasi-steady-state condition (3) must also be fulfilled for the compounds within the exchange compartment, but a defined selection of those can themselves be exported/imported to/from the environment. Accordingly, the stoichiometric matrix $\mathbf{N^c}$ of the community model reads as follows:

$$\mathbf{N}^c = \begin{bmatrix} \mathbf{N_1} & \cdots & 0 & 0 \\ \vdots & \ddots & \vdots & 0 \\ 0 & \cdots & \mathbf{N_n} & 0 \\ \mathbf{E_{1,ex}} & \cdots & \mathbf{E_{n,ex}} & \mathbf{E_c} \end{bmatrix} \qquad (7)$$

$\mathbf{E_{1,ex}} \ldots \mathbf{E_{n,ex}}$ represent the stoichiometries of the exported/imported metabolites from strains $1 \ldots n$

to/from the exchange compartment and $\mathbf{E_c}$ describes the exchange of metabolites between the exchange compartment and the environment. The steady-state constraint (3) for the community now reads:

$$\mathbf{N^c r^c} = \mathbf{0} \qquad (8)$$

## Inclusion of thermodynamics

Several FBA-based modeling techniques incorporate thermodynamic constraints, which reduce the solution space by removing thermodynamically infeasible or unrealistic pathways [32–36]. An example for the usage of thermodynamics-including FBA-based methods in a community model is the analysis of a methanogenic consortium of *S. fumaroxidans* and *M. hungatei* [37]. Thermodynamics-based methods are usually based on the Gibbs free energy $\Delta_r G'$ of (bio)chemical reactions and the fact that the net rate of a reaction can only be positive if its $\Delta_r G'$ is negative. Generally, the $\Delta_r G'_i$ of reaction $i$ depends on the concentrations of the participating reactants and the reaction-specific standard change in Gibbs free energy $\Delta_r G'^0$

[38]:

$$\Delta_r G_i' = \Delta_r G_i'^0 + RT \cdot (\tilde{\mathbf{N}}_{.,i})^{\mathbf{T}} \cdot \mathbf{x} \tag{9}$$

$R$ is the gas constant, $T$ the temperature and $\mathbf{x}$ the logarithmized metabolite concentration vector and $(\tilde{\mathbf{N}}_{.,i})^{\mathbf{T}}$ the transposed $i$-th column (reaction) of the extended stoichiometric matrix $\tilde{\mathbf{N}}$ with explicitly included external metabolites. In the following, we do not distinguish between community and single-strain models, hence, the stoichiometric matrices and reaction rate vectors may also refer to the community versions $\mathbf{N^c}$ and $\mathbf{r^c}$. As a quantitative measure of how thermodynamically feasible a specific reaction $i$ is under given metabolite concentrations, the thermodynamic driving force $f_i$ can be defined, which simply is the negative value of the Gibbs free energy of that reaction:

$$f_i = -\Delta_r G_i' \tag{10}$$

This reaction-centric concept was extended to quantify the thermodynamically feasibility of an entire pathway by introducing the concept of the max-min driving force (MDF) [30]. With a given range of feasible metabolite concentrations, the MDF is the maximal value B such that all reactions of the pathway have a driving force of at least B. The MDF is thus the maximal driving force an entire pathway can reach. The original MDF formulation only allows the computation of the MDF for a given pathway. In order to find, within a genome-scale net-work, a pathway with maximal MDF that fulfills certain constraints (e.g. with a given minimal yield of a certain target product) the OptMDFpathway method was developed [31]. As it is fundamental for the ASTHERISC algorithm presented herein we summarize its basic features.

Binary variables $z_i \in \{0,1\}$ are introduced for each reaction $i$ of the model and we constrain it to be 1 if the associated reaction flux $r_i$ is non-zero ($\beta_i$ is the upper bound for $r_i$):

$$r_i \leq z_i \cdot \beta_i \tag{11}$$

The logarithmized metabolite concentration values in $\mathbf{x}$ are constrained by physiologically meaningful lower ($\mathbf{c_{min}}$) and upper bounds ($\mathbf{c_{max}}$) for the respective concentrations:

$$\ln(\mathbf{c_{min}}) \leq \mathbf{x} \leq \ln(\mathbf{c_{max}}) \tag{12}$$

Sometimes constraints on certain *ratios* of metabolite concentrations are included (e.g. for cofactors such as NADH and NAD). For a fixed ratio $h = \frac{c_i}{c_j}$ of two metabolites $i$ and $j$ we can write

$$x_i - x_j = \ln(h) \tag{13}$$

and a feasible *range* of a metabolite concentration ratio can be expressed by the following two constraints ($h_{min}$ and $h_{max}$ representing the minimal and maximal ratio, respectively):

$$x_i - x_j \geq \ln(h_{min}) \tag{14}$$

$$x_i - x_j \leq \ln(h_{max}) \tag{15}$$

In a preprocessing step, the minimal and maximal possible driving forces $f_{i,min}$ and $f_{i,max}$ under the given concentration ranges (12)-(15) are determined for each reaction $i$. With $K$ we

denote the maximal possible driving force over all reactions

$$K = \max(f_{i,max}) \qquad (16)$$

Next, we introduce $B$ as a lower bound for the driving force of all active reactions:

$$f_i + (1 - z_i) \cdot K \geq B \qquad (17)$$

Finally, maximization of $B$ then yields the optimal (maximal) MDF, abbreviated with OptMDF, for the given constraints:

$$\underset{x,r,B}{Maximize}\ B \qquad (18)$$

$$s.t.\ \text{Eqs (6)–(17)}$$

The defined OptMDF optimization problem combines binary and continuous variables in linear constraints and thus represents a mixed-integer linear program (MILP).

## ASTHERISC algorithm

ASTHERISC (Algorithmic Search of THERmodynamic advantages in Single-species Communities) searches for multi-strain communities that compared to solutions with a single strain, improve the thermodynamic driving force of product synthesis. The algorithm starts with building a community model with $n$ (initially identical) strains of a given species. Standard exchange metabolites (e.g. substrates, nutrients, oxygen, carbon dioxide, common fermentation products such as acetate, ethanol etc.) can be specified, which can be exchanged by the strains with the exchange compartment via *standard exchange reactions* and exchange reactions between the exchange compartment and the environment are added as well to allow uptake/release of these metabolites from/to the environment. In addition, to allow the exchange of (intermediate) metabolites between the strains in the community, we add, to each internal metabolite deemed to be eligible for exchange between the strains, a reversible transport reaction which can import or export the metabolite from/to the exchange compartment. Accordingly, we distinguish *standard exchanges* as introduced above, from *extra exchanges* which allow metabolic interactions between the strains but, in contrast to standard exchanges, no exchange with the environment. The standard as well as the extra exchanges can be specifically defined for every application. Furthermore, when designing a community for the production of a given target metabolite, an export reaction from the exchange compartment to the environment is temporarily added to allow its net production. ASTHERISC searches then for optimal pathways that traverse between the strains to maximize the MDF for synthesizing the target product with a given minimum product yield. The optimal pathway will involve individual (but possibly overlapping) sets of reactions and metabolite concentrations in the different strains.

With this model setup, ASTHERISC employs the OptMDFpathway MILP and slightly modified variants thereof with additional constraints or/and an altered objective function. These variants are described in the following.

1. OptMDF: the original OptMDFpathway algorithm as described above

2. OptMDF2: limited number of (extra) exchanges

If $EX$ contains the indices of the extra exchange reactions, an upper bound $c$ for the number of active extra exchanges can be demanded by adding the following constraint to the OptMDF MILP:

$$\sum_{i \in EX} z_i \leq c. \qquad (19)$$

3. OptMDF3: adding a constraint demanding a lower bound for the MDF

In this variant, we take OptMDF2 and drop the objective function given in Eq (18) and include instead a constraint demanding a minimum value $\gamma$ for the MDF $B$:

$$B \geq \gamma. \tag{20}$$

Note that OptMDF3 just searches for a feasible solution. It returns infeasible if no solution with the demanded minimal MDF exists.

4. OptMDF4: minimize the overall flux

OptMDF4 uses OptMDF3 together with the objective of minimizing the sum of absolute fluxes:

$$Minimize \sum_{i=1}^{q} r_i \tag{21}$$

Note that the OptMDF algorithm splits reversible reactions internally into two irreversible ones ensuring that all rates are non-negative and that (21) really maximizes the sum of absolute fluxes.

5. OptMDF5: searching for minimal/maximal metabolite concentrations

Based on OptMDF3, this variant first maximizes and then minimizes the (logarithmized) concentration of a given metabolite $i$:

$$Maximize \; x_i \tag{22}$$

$$Minimize \; x_i \tag{23}$$

This *concentration variability analysis* is done for all metabolites and determines the feasible metabolite concentrations under the given constraints (including Eq (20)).

The pseudo code of the ASTHERISC algorithm is given in Fig 4. For each (target) metabolite to be tested by ASTHERISC as target product, the ASTHERISC algorithm seeks to find a flux vector and associated metabolite concentrations in the community model such that (a) the community reaches a higher OptMDF than the single-strain solution under (b) the constraint of a minimal defined target product yield. Accordingly, in the first step of the ASTHERISC algorithm, the maximal target product yield, achievable without thermodynamic constraints, is computed for the single-strain model in mol/mol (technically, all single-strain calculations are done via the community model in which only one single strain has unblocked default and target metabolite exchanges) and a specified minimal fraction of this maximum yield must be reached in all subsequent community model calculations. Next, the OptMDF is computed in the single strain and then for multiple strains in the community; in the latter case, an upper bound for the number of active extra exchange reactions can be specified (step 2). If the OptMDF in the multi-strain community model is higher than in the single-strain model, the next steps are performed to characterize the solution (otherwise the algorithm continues with the next target metabolite). Steps 3 and 4 determine the flux vector in the community model that has the minimal flux (sum) under the maximal product yield possible with the found OptMDF. This step removes unnecessary and thermodynamically irrelevant reactions and the obtained minimal flux vector thus indicates the truly required reactions in the MDF-optimal community solution. This flux vector can be optionally further analyzed to determine the feasible metabolite concentration ranges for the optimal community solution (step 5) and to identify thermodynamic bottleneck reactions (step 6) in a single-strain representation of the community solution. As in [31], thermodynamic bottlenecks are defined as those reactions

```
# Set parameters
set singleStrainModel    # the model of the considered (single) species
set numStrains           # number of strains in the single-species community model
set minimalMdfAdvantage  # The minimal necessary MDF advantage for a community
set minimalYieldFactor   # Demanded minimum product yield (as fraction of the maximal
                         # product yield) for a target metabolite
set maxExchanges         # Maximal allowed number of extra exchanges
set targetMetabolites    # A list of indices of target metabolites (the products)

# Build the multi-strain community model with numStrains many strains and add the extra
# exchanges to each strain. This generation of the community model is performed by
# the CommModelPy package, while all other steps are performed by the ASTHERISC package.
communityModel = buildCommunityModel(singleStrainModel, numStrains)

# Loop through each (producible) target metabolite
for targetMetabolite in targetMetabolites:

    # Step 0: Allow (temporarily) an export of the given target metabolite to the
    # environment in the single-strain and in the multi-strain community model.
    singleStrainModelTargetMet= getSingleStrainWithTarget(singleStrainModel, targetMetabolite)

    communityModelTargetMet = getCommunityWithTarget(communityModel, targetMetabolite)

    # Step 1: compute maximal target product yield in single-strain model
    maxYieldSingleStrain = calculateMaxYield(singleStrainModelTargetMet)
    minimalRequiredYield = maxYieldSingleStrain * minimalYieldFactor

    # Step 2: compute maximal MDF (OptMDF) with demanded minimal product yield in both models
    optMdfSingleStrain = OptMDF(singleStrainModelTargetMet, minimalRequiredYield)
    optMdfCommunity = OptMDF2(communityModelTargetMet, minimalRequiredYield, maxExchanges)
    if (optMdfCommunity < optMdfSingleStrain + minimalMdfAdvantage):
        continue  # communityModel not beneficial → try the next targetMetabolite

    # Step 3: determine max product yield at max MDF for community
    communityMaxYield@OptMDF = MaxYield@MaxMDFUsingOptMDF3(communityModelTargetMet, …
            maxExchanges, optMdfCommunity)

    # Step 4: determine min flux vector at max yield at max MDF for community
    communityMinFluxVector@MaxYield@OptMDF = OptMDF4(communityModelTargetMet,…
        communityMaxYield@OptMDF, maxExchanges, optMdfCommunity)
    communityActiveReactions@MinFlux@MaxYield@OptMDF =
        activeReactions(communityMinFluxVector@MaxYield@OptMDF)

    # Step 5 (optional): determine feasible metabolite ranges at min flux vector at
    # max yield at max MDF for community
    metaboliteRanges = calculateMetaboliteRangesUsingOptMDF5(communityModelTargetMet,…
            communityActiveReactions@MinFlux@MaxYield@OptMDF)

    # Step 6 (optional): determine thermodynamic bottlenecks of the community via the
    # single-strain model with active reactions as in the community at max yield and OptMDF
    bottlenecks = calculateBottlenecks(singleStrainModelTargetMet,…
        communityActiveReactions@MinFlux@MaxYield@OptMdf)
```

**Fig 4. Pseudo-code of the ASTHERISC algorithm.** For detailed explanations see text.

where a change of the standard Gibbs free energy alone is enough to further increase the OptMDF value.

Our implementation of the algorithm uses, for some MILPs, indirect approximation steps as they turned out to be advantageous and more robust than direct MILP optimizations in

**Fig 5. Schematic overview of the combined usage of CommModelPy and the ASTHERISC package.** Orange boxes stand for user settings, green boxes for generated or given data files, red boxes for primary program package dependencies, and blue boxes for the programs themselves.

larger models. For example, direct optimization of MDF with the standard OptMDFpathway MILP algorithm may require several hours to days in a genome-scale community model, even when using a high-performance computer cluster. However, approximating the OptMDF with iterative application of OptMDF3 (testing of feasibility with a fixed MDF level) may run in several minutes. This can be used to approximate the OptMDF value by iterative refinement of this value for which a feasible solution can be found (the approximate algorithms stops when a demanded precision level has been reached). The same is done for maximization of product yields under a given optimal MDF constraint.

## Implementation

The generation of community models in the described forms as well as the application of the introduced ASTHERISC algorithm are implemented in two separate packages, the CommModelPy and the ASTHERISC package (Fig 5).

**CommModelPy.** The CommModelPy package can be used to compile multi-strain community models as described in this study, but it also supports construction of multi-species community models under balanced growth. While other Python-based program packages for the generation of communities have been published (including CarveMe [39] and micom [40]), CommModelPy can create community models with a format required for the ASTHERISC package (e.g. fixed species fractions and simulations without growth are allowed). CommModelPy is written in Python and based on the module cobrapy [41]. Among others it supports the SBML format [42]. Herein we used it with Python in version 3.7 and cobrapy 0.17.1. CommModelPy is free and open-source and can be retrieved via its GitHub repository at:

https://github.com/klamt-lab/CommModelPy

The repository also includes a manual and, as a usage example, the CommModelPy-based scripts for the generation of all multi-strain community models used in this study.

## ASTHERISC package

The actual ASTHERISC algorithm with all its subroutines has been implemented in the form of the ASTHERISC package. As input, it takes a CommModelPy-generated multi-strain community model from an SBML file and a list of reaction-associated $\Delta_r G'^0$ values from a JSON file. As output, it generates detailed text reports for which target metabolites OptMDF advantages could be found in a community. The package is written in MATLAB and uses API functions from *CellNetAnalyzer* [43] and the IBM CPLEX solver. Herein, it was used with MATLAB 2018a, *CellNetAnalyzer* 2019.2 and CPLEX 12.9. The ASTHERISC package is free and open-source and can be found under its GitHub repository:

https://github.com/klamt-lab/astheriscPackage

This repository includes all scripts and detailed report files of the ASTHERISC-based analyses performed in this study.

## Results

### Application of ASTHERISC to a small example model

We first illustrate our developed ASTHERISC framework with the toy model given in Fig 2. CommModelPy was used to build a community model. As feasible metabolite ranges, we assumed 1 M-10 M for all metabolites, except for the external product ($P_{ex}$) where we demand a narrower range of 5M-10M so that the product can be synthesized also under higher external product concentrations (see legend of Fig 2). The application of ASTHERISC successfully identified the shown community solution in Fig 2, which improves the maximal thermodynamic driving force as quantified by the OptMDF value. With the single strain, the calculated OptMDF of the pathway from external substrate to external product had a negative value of -0.19 kJ/mol and would thus be infeasible. With the found two-strain community solution, the OptMDF had a positive value of around 0.46 kJ/mol. Hence, in this scenario, the OptMDF is not only increased; the sign change indicates that the community makes product synthesis feasible at all.

The detailed report generated by ASTHERISC for this example is shown in S3 Text. One important information provided in this report is which metabolite concentrations differ in the two strains in the found community solution indicating where a separation in two strains helps to overcome thermodynamic bottlenecks. To obtain the OptMDF of 0.46 kJ/mol in the community solution, X must have a lower concentration between 1 M and 1.37 M in strain 1 and a higher concentration range between 3.65 M and 10 M in strain 2. This explains why the two reactions $R_2$ and $R_5$ had to be separated into strains. Together with the P-exporting reaction $R_6$, which has a low maximal driving force due to the constraint of a high external product ($P_{ex}$) concentration, these two reactions were also identified by ASTHERISC as the sole bottleneck reactions of the single-strain case.

### Applying ASTHERISC to multi-strain community models of *E. coli*

As a practically relevant case, we apply ASTHERISC to identify multi-strain communities of on *E. coli* that could potentially improve the thermodynamic driving force of the production of metabolites. In total, we considered three different community models. Two of them, *ecolicore2double* and *ecolicore2triple*, are based on the *EColiCore2* model [44] (a reduced version of the genome-scale model *i*JO1366 [45] containing 499 reactions and 486 metabolites).

*ecolicore2double* contains two and *ecolicore2triple* three duplicates of the *EColiCore2* network as strains in the community model. The third community model, *iML1515double*, contains two copies of the recently published genome-scale network *i*ML1515 [46]. A detailed description of the construction of the community models, the assignment/computation of $\Delta_r G'^0$, the used standard exchange reactions and other constraints can be found in S2 Text. In short, as the main source of $\Delta_r G'^0$, we employed the Python-based eQuilibrator API, which uses the component contribution method [47]. Reactions, for which no $\Delta_r G'^0$ could be found or calculated (93 reactions (18%) in *EColiCore2* and 646 (24%) in *i*ML1515), were blocked in the model (rate fixed to zero) to avoid the calculation of thermodynamically infeasible solutions when leaving these $\Delta_r G'^0$ unconstrained. Glucose was used as substrate and its uptake rate was fixed to 1 mmol/(gDW·h) (this is only made for technical reasons; it does not change the results since all solutions are scalable). No other flux bounds were set and since growth was not considered this allows use of identical biomass fractions for all involved strain (50% in two-strain communities and 33% in three-strain communities). The three community models were built with CommModelPy and can be found as SBML files in the mentioned GitHub repositories.

With the reduced set of active reactions (due to some missing $\Delta_r G'^0$) and with a demanded minimal product yield of $10^{-6}$ (to prevent potential numeric problems), there were 161 producible target metabolites in the *EColiCore2*-related community models and 254 in *iML1515*double. For each of the producible target metabolites, we considered in each of the three community models 8 different scenarios resulting from four different yield thresholds (40%, 60%, 80% and 98% of the maximal yield of the target metabolite) and two different bounds for the maximal number of (extra) exchange reactions (an infinite or a maximal number of 9 extra exchanges). Hence, 24 scenarios are considered in total. To exclude solutions with just marginal MDF improvements, we demanded a minimal community MDF advantage of 0.2 kJ/mol and a positive OptMDF value was demanded for the community (not for an associated single-strain) solution in order to get thermodynamically feasible results.

For all 24 scenarios, ASTHERISC could find a wide range of target metabolites where a community was able to provide a pathway with a higher OptMDF than a single strain (for a complete listing of all results see the report files in the ASTHERISC package's repository). The percentage of producible metabolites for which an advantage could be found in the community ranged from 15.75% up to 40.55% (Table 1). In most cases, as expected, limiting the maximal extra exchanges to 9 reduces the number of found improvements compared to the case with unlimited exchanges. Since the computation was stopped for several metabolites due to MILP timeouts (especially in the genome-scale community model), the percentages given for the different scenarios should be seen as lower bound of possible improvement percentages.

The OptMDF advantages of community solutions compared to single-strain solutions ranged from 0.22 kJ/mol (just over the demanded minimal MDF advantage of 0.2 kJ/mol) up to

**Table 1. Percentage (and absolute number) of all producible target metabolites for which ASTHERISC found a higher optimal MDF in a multi-strain community compared to a single-strain model.** Note that only those communities were considered, where the MDF was at least 0.2 kJ/mol higher than in the single-strain model.

| Model (number of allowed extra exchanges) | Demanded minimum product yield (% of maximum yield) | | | |
| --- | --- | --- | --- | --- |
| | **40%** | **60%** | **80%** | **98%** |
| *ecolicore2double* (9) | 22.36% (36) | 21.74% (35) | 21.12% (34) | 21.12% (34) |
| *ecolicore2double* (infinite) | 23.6% (38) | 35.4% (57) | 27.95% (45) | 31.06% (50) |
| *ecolicore2triple* (9) | 21.12% (34) | 19.88% (32) | 17.39% (28) | 19.25% (31) |
| *ecolicore2triple* (infinite) | 19.25% (31) | 31.06% (50) | 27.33% (44) | 29.81% (48) |
| *iML1515double* (9) | 17.72% (45) | 15.75% (40) | 21.26% (54) | 25.59% (65) |
| *iML1515double* (infinite) | 18.5% (47) | 20.08% (51) | 25.59% (65) | 40.55% (103) |

**Table 2. Statistics about OptMDF advantages (in kJ/mol) with communities for each of the 24 scenarios.** The given numbers stand for the minimal/mean/maximal OptMDF advantage.

| Model (number of allowed extra exchanges) | Demanded minimum product yield (% of maximum yield) | | | |
|---|---|---|---|---|
| | 40% | 60% | 80% | 98% |
| ecolicore2double (9) | 0.32/0.94/2.76 | 0.2/0.64/3.13 | 0.29/0.67/3.24 | 0.27/2.0/5.49 |
| ecolicore2double (infinite) | 0.32/1.29/3.42 | 0.26/1.02/5.07 | 0.22/1.04/5.07 | 0.23/1.77/6.28 |
| ecolicore2triple (9) | 0.32/0.83/2.76 | 0.22/0.63/3.03 | 0.29/0.7/3.24 | 0.5/1.85/5.49 |
| ecolicore2triple (infinite) | 0.29/1.42/3.42 | 0.26/1.09/4.71 | 0.22/0.93/4.71 | 0.23/1.81/6.28 |
| iML1515double (9) | 0.26/0.92/2.95 | 0.33/1.0/2.79 | 0.23/0.93/2.66 | 0.22/1.58/5.55 |
| iML1515double (infinite) | 0.26/0.97/3.04 | 0.26/0.91/2.79 | 0.3/0.91/2.64 | 0.22/1.94/6.04 |

7.26 kJ/mol, while the mean MDF advantages range from 0.64 kJ/mol to 2.0 kJ/mol for *ecolicore2double* and *ecolicore2triple* and from 0.91 kJ/mol to 1.94 kJ/mol for *iML1515double* (Table 2). Interestingly, the highest MDF improvements occur when high product yields are demanded. This indicates that low-yield pathways in the single strain might have high MDFs which are more difficult to improve by a community solution. We also found some solutions in which the MDF of the single strain is negative and in the community positive, hence, where product synthesis becomes thermodynamically feasible at all in the community (similar as in the toy model scenario in Fig 2). This was the case for 7 metabolites in all runs with a yield threshold of 98% with *ecolicore2double* and *ecolicore2triple*, and for 3 metabolites in all runs with *iML1515double*.

Next, we analyzed how many metabolite exchanges were used in the found community solutions (Table 3). For the scenario with a limited number of (extra) exchanges, it should be noted that a maximum of 9 active exchange reactions in a community with two strains correspond to an exchange of maximal 4 metabolites between the two strains: 4 exchange reactions from strain 1 for export/import of metabolites and 4 reactions of strain 2 for import/export of the same 4 metabolites plus the export of the target product by one of the two strains. The number of used metabolite exchanges per target metabolite solution ranged from 1 (in those cases, default metabolite exchanges were typically used between the strains which are not counted as extra exchanges) up to 51.

Some metabolites occur with high frequency as exchange metabolites in the found solutions. For example, glycerol3-phosphate, (glyc3p), fructose-6-phosphate (f6p) and oxaloacetate (oaa) are top-ranked exchange metabolites in *ecolicore2double* and *ecolicore2triple* while dihydroxyacetone (dha), L-aspartate (asp__L) and acetaldehyde (acald) are often contained in solutions for *iML1515double*. Most of the top-ranked exchange metabolites are involved in the glycolysis or in pathways directly connected to glycolysis indicating central thermodynamic bottlenecks relevant for the production of many target metabolites.

**Table 3. Statistics about the number of used extra exchanges in solutions with a community OptMDF advantage for each of the 24 scenarios analyzed.** The given numbers stand for the minimal/mean/maximal number of used extra exchanges.

| Model (number of allowed extra exchanges) | Demanded minimum product yield (% of maximum yield) | | | |
|---|---|---|---|---|
| | 40% | 60% | 80% | 98% |
| ecolicore2double (9) | 3/5.39/8 | 4/6.34/7 | 4/6.41/9 | 3/6.03/9 |
| ecolicore2double (infinite) | 3/16.66/29 | 7/15.88/31 | 6/13.76/29 | 8/15.88/41 |
| ecolicore2triple (9) | 3/4.47/7 | 3/5.88/7 | 4/5.89/7 | 3/5.29/9 |
| ecolicore2triple (infinite) | 3/18.13/38 | 3/14.8/36 | 5/13.36/34 | 4/16.83/51 |
| iML1515double (9) | 1/5.18/7 | 3/5.05/9 | 1/5.11/9 | 2/5.14/9 |
| iML1515double (infinite) | 3/15.28/33 | 3/14.16/27 | 3/14.11/37 | 3/14.34/28 |

Coinciding with the dominant extra exchange reactions, a few discrete MDF values are often dominating for each scenario. In particular, if target metabolites are closely located in the metabolic network, then their corresponding community solutions have often the same associated OptMDF value as they share the same metabolic bottleneck for their production. Some of these dominant MDF values occurred in all analyzed models. This might indicate that a small set of central bottleneck reactions is thermodynamically constraining a large set of possible solutions.

## Community solution example

To discuss one representative solution as an example, we selected the production of 3-Deoxy-D-manno-octulosonate 8-phosphate (kdo8p) in *ecolicore2double* with a demanded minimal product yield of 98% of the maximum yield (0.74775 mol kdo8p per mol glucose) and with an allowed maximum of 9 extra exchanges. The last step of kdo8p synthesis is catalyzed by the kdo8p synthase (KDOPS), which has been evaluated as a potential antibiotics target [48]. In the following, the two designed strains within the *ecolicore2double* community model will be denoted by "ecoli1" and "ecoli2", respectively. The report delivered by ASTHERISC for kdo8p can be found in S4 Text.

With the given constraints, an OptMDF of 2.27 kJ/mol can be reached for the production of kdo8p from glucose with a single strain. In the community solution found by ASTHERISC, the MDF value can be increased by 26% up to 2.87 kJ/mol reaching a yield of 0.73311 mol/mol, which is just 0.04% higher than the demanded minimal yield. While the single-strain solution uses 42 reactions (thereof 14 standard exchange reactions for glucose, water, etc.), the community requires 66 reactions, thereof 22 exchange reactions, 28 internal reactions in ecoli1 and 16 reactions in ecoli2. 17 of the 22 exchange reactions are standard exchanges and four extra exchange reactions are used for metabolite exchanges between ecoli1 and ecoli2 (Fig 6): two for the exchange of dihydroxyacetone (dha) from ecoli1 to ecoli2, two for the exchange of 6-Phospho-D-gluconate (6pgc) from ecoli2 to ecoli1 and one for the secretion of kdo8p to the environment.

To understand why this community solution leads to an MDF advantage, it is helpful to look at the three thermodynamic bottleneck reactions for kdo8p synthesis in the MDF-optimal single-strain solution (Fig 6). These three connected reactions (shown in red in Fig 6), which all occur in the upper glycolysis, are (1) the fructose-bisphosphate aldolase reaction (FBA; $\Delta_r G'^0$ = 22.4 kJ/mol) producing glyceraldehyde 3-phosphate (g3p) and dihydroxyacetone phosphate (dhap) from D-fructose 1,6-bisphosphate (fdp), (2) the triose-phosphate isomerase reaction (TPI; $\Delta_r G'^0$ = 5.6 kJ/mol) converting dhap to g3p, and (3) the glyceraldehyde-3-phosphate dehydrogenase reaction (GAPD; $\Delta_r G'^0$ = 4.6 kJ/mol) where g3p is converted to 3-phospho-D-glycerol phosphate (13dpg). As already pointed out in [30] and [31], these three reactions form a distributed thermodynamic bottleneck as they all have unfavorable thermodynamics to operate in forward direction (positive $\Delta_r G'^0$) and conflicting needs regarding the g3p concentration since the FBA and TPI reactions thermodynamically benefit from a lower and the GAPD reaction from a higher g3p concentration. There is also a fourth associated bottleneck reaction, the symporter-assisted uptake of phosphate into the cytosol (PIt2rpp) with a $\Delta_r G'^0$ of 0 kJ/mol, which delivers the phosphate needed in the GAPD reaction. For the sake of simplicity, it was not included in Fig 6. The optimal adjustment of metabolite concentrations in the single strain allows a maximum MDF value of 2.27 kJ/mol and changing the concentration, e.g. of g3p, will inevitably lead to an increase of the $\Delta_r G'$ of at least one reaction and thus to a decrease of the MDF.

In the community solution, the bottleneck is resolved as follows (Fig 6): In order to produce kdo8p through the reaction KDOPS (kdo8p synthase), both phosphoenolpyruvate (pep) and

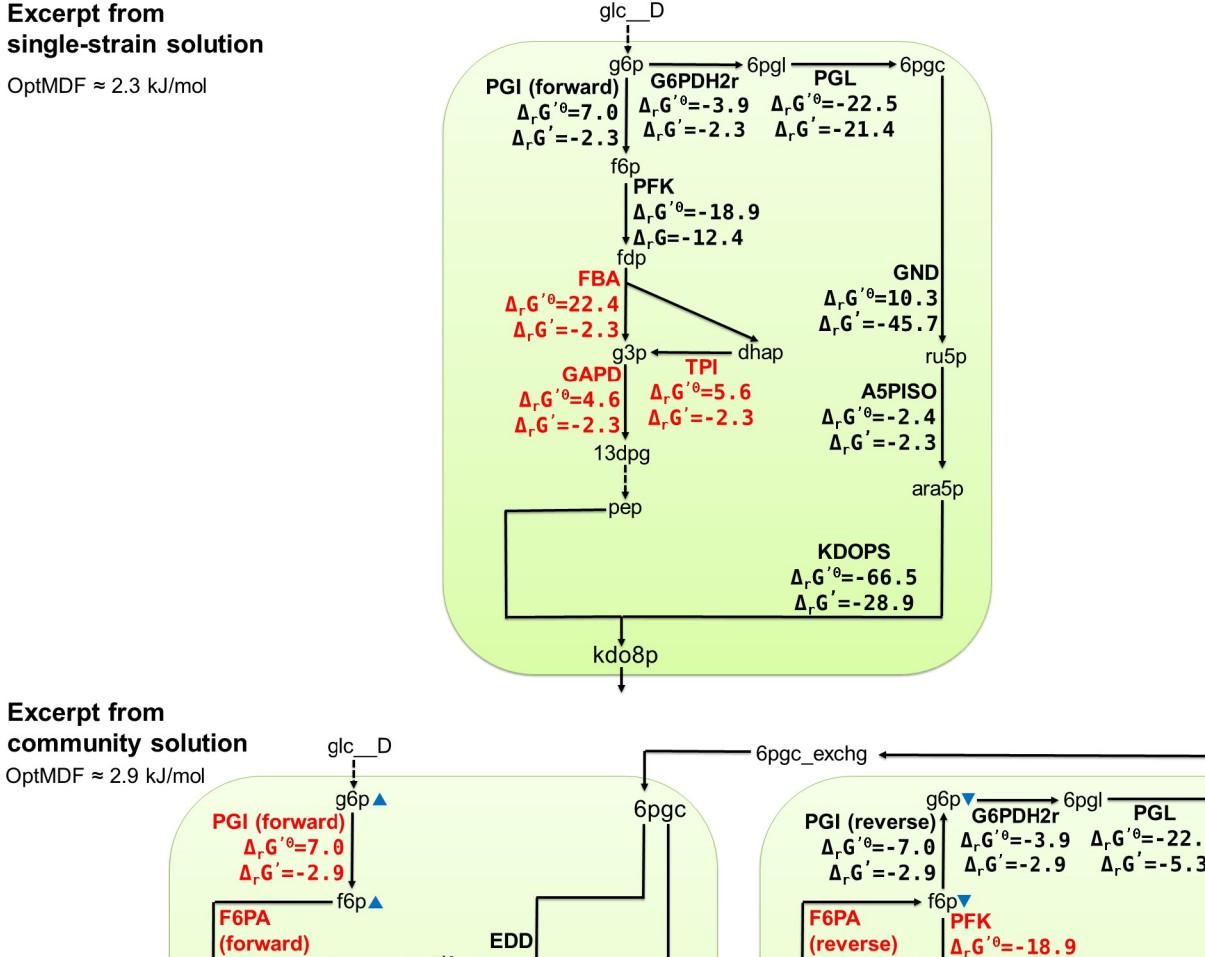

**Fig 6. Excerpt of selected central reactions of the MDF-optimal single-species and community solution for kdo8p synthesis.** Black arrows indicate active reactions, dashed arrows indicate a sequence of active reactions, and the blue triangles indicate increased or decreased metabolite concentrations of a strain in the community relative to the other strain. The shown $\Delta_r G'$ and $\Delta_r G'^0$ values have all unit kJ/mol. The $\Delta_r G'$ values are taken from the specific MDF-optimal solution delivered by ASTHERISC. The red $\Delta_r G'$ values indicate thermodynamic bottlenecks, i.e., reactions whose $\Delta_r G'$ is fixed under the optimal MDF (and corresponds to the negative value of the OptMDF) All black $\Delta_r G'$ values are variable under the given OptMDF. All reaction and metabolite identifiers are based on the definitions in the BiGG database [49].

D-arabinose 5-phosphate (ara5p) are needed as substrate. In the single strain solution, pep is produced using the glycolysis, thus requiring the bottleneck reactions FBA, TPI and GAPD. ara5p is produced through reactions which do not pose a bottleneck, using a pathway via 6-phospho-D-gluconate (6pgc). In the community solution, the bottleneck triangle of the FBA, TPI and GAPD reactions is resolved by separating the occurrence of the g3p-using GAPD reaction into ecoli1 and the g3p-producing FBA and TPI reactions into ecoli2, thus omitting the conflicting needs of g3p. This is achieved in the following way: (1) ecoli1 concentrates on pep synthesis, thereby omitting parts of the upper glycolysis by using the fructose 6-phosphate aldolase reaction (F6PA), which has f6p as a substrate and dihydroxyacetone (dha) and g3p as products. g3p is further processed using GAPD, and dha is secreted and taken up by ecoli2 for further processing (see below). With a higher concentration of f6p, a higher and thermodynamically more favorable concentration (compared to the single-strain solution) of g3p is achieved in ecoli1, so that the $\Delta_r G'$ value of the GAPD reaction can be lowered (and thus the MDF be increased) in the community solution. Furthermore, a higher f6p concentration also requires an increased D-glucose 6-phosphate (g6p) concentration since f6p is produced from it using the glucose-6-phosphate isomerase reaction (PGI). (2) ecoli2 concentrates on the production of 6pgc from dha. In a first step, the reverse F6PA reaction is used to produce f6p from dha and g3p. While dha is provided from ecoli1, the required g3p is recycled from a portion of the produced f6p via the g3p-producing reactions FBA and TPI. Due to the very negative $\Delta_r G'^0$ of the reverse F6PA reaction, low concentrations of dha and g3p can be used in ecoli2 resulting in a more negative $\Delta_r G'$ of the FBA and TPI reactions. The second half of f6p is used to synthesize 6pgc via the (reverse) PGI, G6PDH2r and PGL reactions. In order to make all these reactions work with low $\Delta_r G'$ (resulting in high MDF), the concentrations of dha, f6p, g6p and, as mentioned before, of g3p are reduced compared to ecoli1. The three reactions on the pathway from f6p to 6pgc still reach sufficient thermodynamic driving force because of the very negative $\Delta_r G'^0$ of these reactions. (3) The 6pgc produced by ecoli2 is then secreted and taken up by ecoli1and further processed via two pathways to synthesize pep and D-arabinose 5-phosphate (ara5p), the two precursors of kdo8p. pep is produced via g3p and pyruvate (pyr) using Entner-Doudoroff pathway reactions, including the 6-phosphogluconate dehydratase reaction (EDD) with a very low $\Delta_r G'^0$ of -42.8, which allows the concentrations of the affected metabolites to be more variable in order to obtain a high MDF. Note that, in ecoli1, some additional amount of pep is produced via glycolytic reactions from g3p using, for example, the GAPD reaction (Fig 6). The other precursor ara5p is synthesized via the two reactions GND and A5PISO. Finally, ara5p and pep are used by KDOPS to produce kdo8p, which is then secreted into the environment. It has to be mentioned that more reactions are involved in the single-strain as well as in the multi-strain solution to obtain a stoichiometrically balanced flux (not displayed in Fig 6). These reactions are required, for example, to balance cofactors (ATP, NAD(P)H) produced or consumed along the sketched pathways, however, they are not critical for the respective MDF.

Overall, the community solution follows the basic principle of the toy model (Fig 2): through a suitable partitioning (and partially complementation) of the pathway from glucose to the target product kdo8p in the two strains, critical metabolite concentrations (in this example, dha, f6p, g3p and gp6 as indicated in Fig 6) can be maintained with different levels in ecoli1 (high concentrations) and ecoli2 (low concentrations), thereby reaching more negative $\Delta_r G'$ for the bottleneck reactions and thus a higher overall MDF. This can also be seen by the feasible concentrations ranges of the critical metabolites in ecoli1 and ecoli2 which maintain the OptMDF of 2.87 KJ/mol (Table 4). These metabolites, which occur in the bottleneck reactions of the single-strain solution, have no overlapping concentration ranges between ecoli1 and ecoli2 showing that the given community solution indeed cannot be reached in a single

**Table 4. Concentration ranges of critical metabolites in the *ecolicore2double* community solution for kdo8p synthesis with optimal MDF.** The concentration ranges do not overlap between the two strains.

|  | ecoli1 | | ecoli2 | |
|---|---|---|---|---|
|  | **Minimum [mmol/l]** | **Maximum [mmol/l]** | **Minimum [mmol/l]** | **Maximum [mmol/l]** |
| **dha_c** | 0.59826 | 0.59905 | 0.058969 | 0.059048 |
| **f6p_c** | 0.37078 | 0.37128 | 0.001 | 0.0010027 |
| **g3p_c** | 0.32482 | 0.32525 | 0.090047 | 0.090286 |
| **g6p_c** | 19.973 | 20 | 0.0019629 | 0.053238 |

strain. Interestingly, the community solution has in turn new thermodynamic bottlenecks, again involving the reactions FBA, TPI and GAPD. Those might be further resolved by using a community with more than two strains. However, the corresponding kdo8p solution with *ecolicore2triple* does not provide a higher OptMDF in the community, indicating that more than three species might be needed in order to achieve a further MDF advantage.

## Run time

For all calculations we used a high-performance computer cluster comprising, for each of its nodes, 192 GB working memory and two processors each including eight cores with a standard clock rate of 2.1 GHz. Each of the mentioned 24 scenarios was running on its own node. The resulting run times for the scenarios (cumulated for all producible target metabolites) ranged from 1.63 days (*ecolicore2double* with a minimal yield factor of 98% and 9 maximal extra exchanges) up to 4.54 days (*iML1515double* with a minimal yield factor of 80% and 9 maximal extra exchanges).

## Discussion

In this study we presented ASTHERISC, a new algorithm for designing multi-strain microbial communities that maximize the thermodynamic driving force of product synthesis with a given production host. The key methodological approach behind ASTHERISC is to compartmentalize certain segments of the product pathway via different strains and to allow an exchange of intermediates between these strains such that the overall product synthesis pathway through the involved strains may reach a higher thermodynamic driving force than in the single strain. While the set of active reactions in the different strains can overlap, using more than one strain adds degrees of freedom with respect to the achievable MDF because the metabolite concentrations may differ between the strains. In this way, thermodynamic bottlenecks, which may arise when operation of one reaction requires a metabolite with high and operation of another reaction the same metabolite with a low concentration, can be circumvented.

In recent years, microbial consortia and their potential for biotechnological production processes have attracted much attention. However, although several single-species (multistrain) communities with essential metabolic dependencies between the strains have been constructed and analyzed, there are still only few methods available for an automated design of communities with superior properties [5], with FLYCOP [50] and DOLMN [17] as two particular examples. Here, ASTHERISC adds a new class of such algorithms which is driven by a thermodynamic principle. The recently presented DOLMN approach [17], which searches for an optimal partitioning of a species' metabolism when limiting the number of reactions per strain, shares some similarities with ASTHERISC. However, to the best of our knowledge, a design of microbial communities with the goal to optimize the thermodynamic feasibility of production of a target metabolite has not been addressed before.

The application of our algorithm to a metabolic core and a genome-scale model of *E. coli* revealed that there is indeed a significant number of metabolic products where a multi-strain community of *E. coli* may provide higher thermodynamic driving forces. These communities may be an effective way to enhance or even enable the synthesis of certain metabolic products. ASTHERISC, together with the developed software packages, provides an automatable way to identify such beneficial communities consisting of multiple strains of one species. However, while the found solutions demonstrate that higher thermodynamic driving forces may be generated with specifically designed communities, significant effort will be needed for their practical realization. First of all, designing those strains will require the deactivation of certain metabolic reactions (e.g. through gene knockouts) according to the found community solution. Finding suitable knockout strategies that leave only the relevant pathway segments active in the respective strains can be computed by dedicated strain design algorithms, e.g. based on minimal cut sets [51]. Implementing the respective gene knockout strategies may lead to unstable strains or even prevent growth. While the (stoichiometric) feasibility of growth may be demanded when computing the respective intervention strategies, it might be more practical to use two-stage processes, where the strains can first grow before a pure production phase is initiated (e.g. via dynamic metabolic switches [52,53]). This also circumvents possible limitations in the MDF of the production pathway when growth and production are coupled (in our calculations we focused on optimal pathway partitioning for pure product synthesis and did not consider feasibility of growth).

Next, transport of exchange metabolites between the strains must be faciitated (e.g. via overexpression of dedicated transport systems). Transport systems are only available/known for a subset of metabolites and solutions involving only few exchanges may thus be preferred. However, many of the found community solutions for different products use similar sets of exchange metabolites. Hence, once suitable transporter mechanisms have been discovered or implemented for these metabolites, a larger number of strain designs may become feasible. As another aspect for metabolite transport, it was assumed for all exchange reactions that a concentration gradient must exist between the cellular and the external compartment, however, transport costs, e.g. energy consumption to actively transport a metabolite over the membrane, were not explicitly considered (except for transporters already included in the base model). These overhead costs may have adverse effects on product yields and also abolish the MDF advantage of a community solution. To test the sensitivity of the latter when introducing energy costs for metabolite transport, we reran all calculations in a variant of the *ecolicore2double* model, where each introduced (extra) metabolite exchange consumes 1/3 ATP per transported metabolite molecule. Although there are few specific changes in the set of metabolites where an MDF advantage can be found, we did not find larger changes in the overall percentage of metabolites with MDF advantage in a community. In fact, export of the (end) product does then also require ATP in the single-strain solution, which may lead to a change in the structure of the active network possibly favoring a solution in the community. The associated scripts and result files can be found in the already mentioned GitHub repositories of CommModelPy and the ASTHERISC package.

Another issue for realization of a calculated community design is the long-term stability of the community, which may require genetic methods and general principles as developed in [54] and, more recently, [9]. Finally, reaching the computed maximal thermodynamic driving forces for the communities also demands that the metabolite concentrations are within the respective ranges required for the optimal MDF and it is not per se clear whether this will be the case or can be adjusted by suitable selection mechanisms. It thus remains to be shown that the computed communities may constitute an economically relevant enhancement. However, it should be noted that many of the discussed difficulties are relevant for the implementation of any (computed) synthetic community.

The ASTHERISC runs in the example calculations of this study were stable. Potential numerical problems arising due to precision tolerances of the MILP are partially directly prevented or detected and handled by our implementation. For example, "fake community solutions" delivered by the solver, where effectively only one strain is used, are detected and disregarded. It should also be noted that the set computational time limits may affect the results. This can be seen in the reduced MDF advantage percentages with *ecolicore2triple* compared to *ecolicore2double* (Table 1), where the MDF advantage in *ecolicore2triple* should be at least as high as in *ecolicore2double* as the former should implicitly contain all two-strain solutions. This effect arose due to MILP time limits which were more likely to be reached in the larger model. Nevertheless, the runs of the *ecolicore2triple* community model clearly showed that a larger number of strains may increase the overall community advantage. Finally, the calculations with the *iML1515double* model demonstrated that our implementation of the ASTHERISC algorithm can be applied to genome-scale models with a reasonable runtime. Indeed, comparing the found solutions for *ecolicore2double* and *iML1515double* indicates that, for some metabolites, other MDF-optimizing pathways may be found in a genome-scale model, although they may involve more uncommon pathways (with possibly lower capacity) than in the solutions found in the core model.

The results obtained with ASTHERISC also depend on the used settings (e.g. demanded minimum MDF advantage and minimum product yield) and the calculations involve partially uncertain parameters or assumptions (e.g. the metabolite concentration ranges or the $\Delta_r G'^0$ calculated with the eQuilibrator method [47]). However, the calculations can be easily repeated with alternative values if these are considered to be more relevant or evident. Furthermore, the developed CommModelPy and ASTHERISC packages are programmatically independent from the used models and can thus be readily applied for the generation and study of other single-species. Although not yet tested, ASTHERISC could also be used with multi-species communities, for example, to test which product synthesis pathway (with associated metabolite exchanges) in a multi-species community leads to the highest MDF and whether this MDF is superior over a potentially existing single-species pathway. Finally, we also anticipate useful applications of the ASTHERISC method for finding suitable compartmentalization strategies maximizing thermodynamic driving forces in cell-free production systems.

Community design by ASTHERISC exploits the fact that having multiple strains allows adjustment of different metabolite concentrations in the different strains. While ASTHERISC uses this degree of freedom for maximizing the thermodynamic driving force, future work could seek to directly maximize the product synthesis rate in the community by finding a pathway partitioning (again with specific metabolite concentrations in each strain) such that, under consideration of kinetic rate laws, the overall product flux is optimized. Ideally, if a kinetic model of the (central) metabolism would be available one could directly search for such an optimal solution. For example, if we would have a simple kinetic model of the toy network in Fig 2, it would most likely suggest to split the pathway in the same way as was done via the thermodynamic principles to maximize the pathway flux. However, a kinetic approach would also allow the consideration of saturation effects, allosteric regulations (e.g. feedback inhibition of the first reaction by intermediate or end metabolites of the pathway) or of resource (enzyme) allocation constraints. Since predictive kinetic models are often not available, it needs to be investigated whether other approaches requiring less information (e.g. based on $k_{cat}$ or $K_m$ values [55,56]) may already suggest useful solutions.

## Supporting information

**S1 Text. Proof showing that division of labor cannot increase the product synthesis flux compared to a single-strain solution for arbitrary pathways and kinetics (under**

**assumption of constant metabolite concentrations in the strains).**
(PDF)

**S2 Text. Detailed description of configuration of the single-strain models and the derived multi-strain community models of *E. coli*.**
(PDF)

**S3 Text. The ASTHERISC package text report generated for the toy model.**
(PDF)

**S4 Text. The ASTHERISC package text report generated for the discussed community solution example for kdo8p synthesis with *ecolicore2double*.**
(PDF)

## Acknowledgments

We are grateful to Axel von Kamp for his assistance and discussions on using OptMDFpathway in the context of ASTHERISC.

## Author Contributions

**Conceptualization:** Steffen Klamt.

**Data curation:** Pavlos Stephanos Bekiaris.

**Formal analysis:** Pavlos Stephanos Bekiaris.

**Funding acquisition:** Steffen Klamt.

**Investigation:** Pavlos Stephanos Bekiaris.

**Methodology:** Pavlos Stephanos Bekiaris, Steffen Klamt.

**Project administration:** Steffen Klamt.

**Resources:** Steffen Klamt.

**Software:** Pavlos Stephanos Bekiaris.

**Supervision:** Steffen Klamt.

**Validation:** Pavlos Stephanos Bekiaris, Steffen Klamt.

**Visualization:** Pavlos Stephanos Bekiaris, Steffen Klamt.

**Writing – original draft:** Pavlos Stephanos Bekiaris, Steffen Klamt.

**Writing – review & editing:** Pavlos Stephanos Bekiaris, Steffen Klamt.

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
