## [Decision Letter · Decision Letter 0]

17 Mar 2021

Dear Dr. Klamt,

Thank you very much for submitting your manuscript "Designing Microbial Communities to Maximize the Thermodynamic Driving Force for the Production of Chemicals" for consideration at PLOS Computational Biology.

As with all papers reviewed by the journal, your manuscript was reviewed by members of the editorial board and by several independent reviewers. In light of the reviews (below this email), we would like to invite the resubmission of a significantly-revised version that takes into account the reviewers' comments. Please pay particular attention to the points raised with regard of the justification of assumptions that your algorithm makes.

We cannot make any decision about publication until we have seen the revised manuscript and your response to the reviewers' comments. Your revised manuscript is also likely to be sent to reviewers for further evaluation.

Sincerely,

Christoph Kaleta

Associate Editor

PLOS Computational Biology

Daniel Beard

Deputy Editor

PLOS Computational Biology

Reviewer's Responses to Questions

**Comments to the Authors:**

Reviewer #1: The authors present a new method, ASTHERISC, for designing synthetic microbial communities with optimal thermodynamic driving force for the production of a target compound. The method is novel with regard to being the first community design method that accounts for thermodynamics, and I believe it is quite relevant to other people working in this field. However, I think the authors should discuss some limitations in a bit more detail. Please see my comments bellow.

- The theoretical model of division of labor (DoL) assumes that there is no cost associated with the transport of the exchanged metabolic intermediates. For instance, in Figure 1, if one adds a cost term for the transport of M, the DoL model actually has an increased metabolic burden. This issue is only very briefly mentioned in the discussion, but given that it can make the whole concept unfeasible in a real scenario, I think it should be better discussed. For instance, the authors could have tested the sensitivity of the solutions with respect to the addition of an exchange cost (not just a limit on the total number of exchanges).

- Also, the authors add artificial exchange reactions to transport the metabolites between the cytosol and the growth medium, which leads to the exchange of metabolites, like phosphorylated sugars, that in principle cannot be transported outside the cell (glyc3p and f6p are reported to be among the most frequently exchanged). Why not simply use the transporters that are already present in the models ?

- Another aspect which is not discussed is what would happen during the scale-up to very large (i.e. not well-mixed bioreactors). Could the rate of diffusion of the metabolic intermediates become a bottleneck?

- The results presented in Table 1 show the fraction of cases where the optimal MDF is increased. But it is not mentioned how marginal this increase can be. I think the application of ASTHERISC is mainly relevant to the case illustrated in Fig 2, where a pathway with negative OptMDF can be split between species to obtain a positive MDF. How many of the cases presented in table 1 are actually of this kind?

- Why are the metabolite concentrations specified in M (i.e. mol/L)? What is the volume referring to in this case? Is it the cell volume? Shouldn't these values be 1000-fold lower, i.e., in the mM range ?

- I don't understand why the authors opted to block the reactions with unknown dG0's, which is essentially the same as removing them from the models. I think it would be more realistic, and biologically meaningful, to leave them unconstrained, since in reality these reactions would be able to carry flux, potentially deviating flux from the target pathway.

- I didn't see any mention about uncertainty in the dG0 calculations, which is one of the outputs returned by eQuilibrator. Were the uncertainties ignored? Considering that some can be quite large, shouldn't they be included in equation 9 (i.e. by replacing the equality with lower/upper bounds)?

- The algorithm doesn't actually propose any strain design targets to implement the mutant strains. It is only briefly mentioned during the discussion that this is something that must be done after computing the optimal pathway. However, considering that the title of the paper is Designing Microbial Communities (...), I think the authors should elaborate a bit more

on this aspect.

- Small suggestion: all the four tables presented could be more intuitively represented as plots.

Reviewer #2: The comments are also attached as a pdf. It might be easier to read the equations there.

### Peer review of *Designing Microbial Communities to Maximize the Thermodynamic Driving Force for the Production of Chemicals*

The authors present an interesting algorithm for the computational design of single-species communities that maximise the thermodynamic driving force of producing a certain target metabolite. The paper is written in a clear and educational style, which makes it pleasant to read. The algorithm is potentially important, and has novelty for both the modelling of microbial communities and the design of bio-based production. Although I encountered some issues that I think should be resolved, I think the manuscript is fit for publication in Plos Computational Biology after some major revisions.

My main criticism is related to the strong and non-negligible assumptions that the ASTHERISC algorithm is based on. I understand that making these assumptions is necessary: the modelling of microbial communities is difficult, and some assumptions are needed to make computational problems feasible. However, I think the reader should be guided much more in interpreting the consequences of these assumptions. My remarks, ordered per section, can be found below.

#### Remarks Introduction

##### The example described in Figure 1

Although I appreciate the attempt to introduce the concept of DoL in a very basic way, the example model is a bit too basic for my taste. At a first glance, I am left with the idea that the assumptions that are made are necessary to draw the conclusions that the authors want to make. I think this is not the impression that the authors want to leave, because their argument is much more general. Also, the statement “Hence, it is unlikely that a DoL strategy will be advantageous with respect to the metabolic burden …” is repeated a couple of times in the introduction, and I think it is not entirely correct. See my explanation below on how this could be nuanced.

The two assumptions that I find too specific are:

- the saturation functions only depend on the substrate of the reaction, rather than also on the product.

- the same amount of enzymes E1 and E2 is needed for a specific flux

I know that the authors already state that the second assumption does not affect the conclusions, but the reader is still left wondering what the effect of this assumption might be.

I would suggest two edits:

1. I think it is fairly straightforward to write a small mathematical proof that DoL does not give an advantage when the metabolite concentrations are unchanged, i.e., the saturation functions remain the same. This proof can be added to the SI, and can be referred to in this Introduction. I have added an outline for the proof below.

Adding this proof would also allow for a more exact statement. The authors now state: *“Hence, it is unlikely that a DoL strategy will be advantageous with respect to the metabolic burden … ”* But this should actually be something like: *“Hence, it is unlikely that a DoL strategy will be advantageous with respect to the metabolic burden, as long as the kinetic advantage of having different metabolite concentrations in the different strains do not outweigh the added costs of metabolite transport.”* The statement is a bit less strong, but definitely more correct. Also, adjusting this statement would allow the authors to give a preview on the rest of their work, because adding a growth advantage by having different metabolite concentrations is exactly what they are going to do, albeit focusing on thermodynamics and not on kinetics.

2. In this Introduction I would at least remove the first assumption, and just make the saturation functions depend on all metabolite concentrations. This does not change the reasoning at all. Personally, I would also skip the second assumption, but I realize that this is a matter of taste. Generalising this here might complicate matters. It is also not really necessary as long as you can refer to the general proof in the SI.

###### An idea for the proof:

*I wanted to write a quick outline for the proof, but it got a bit more involved than I expected. I hope the proof below is useful for the authors. Please check everything carefully, because I did not.*

One could start by taking a general reaction vector, $v$, that leads to some production of $P$. We can multiply $v$ by a constant and still have the production pathway, and the production rate of $P$ is proportional to the reaction rates in $v$. Let $\\tilde v$ now be rescaled such that $\\tilde v_P = 1$. The enzyme costs of this production pathway can be expressed in terms of some (arbitrary) saturation function $f_i(S,M,P,…)=:f_i(x)$ for each reaction $i$. The enzyme costs for having $v_P=1$ are then

<math xmlns="http://www.w3.org/1998/Math/MathML"><semantics><mrow><munder><mo>∑</mo><mi>i</mi></munder><msub><mi>e</mi><mi>i</mi></msub><mo>=</mo><munder><mo>∑</mo><mi>i</mi></munder><msub><mover><mi>v</mi><mo>~</mo></mover><mi>i</mi></msub><mo>/</mo><mo>(</mo><msub><mi>k</mi><mrow><mtext>cat</mtext><mo>,</mo><mi>i</mi></mrow></msub><msub><mi>f</mi><mi>i</mi></msub><mo>(</mo><mi>x</mi><mo>)</mo><mo>)</mo><mo>.</mo></mrow><annotation encoding="LaTeX">\\sum_i e_i = \\sum_i \\tilde v_i/(k_{\\text{cat},i}f_i(x)).</annotation></semantics></math>

So, given a limited amount of resources ($\\sum_i e_i \\leq C$) we can get a maximal production flux of

<math xmlns="http://www.w3.org/1998/Math/MathML"><semantics><mrow><msub><mi>v</mi><mi>P</mi></msub><mo>=</mo><mfrac><mi>C</mi><mrow><msub><mo>∑</mo><mi>i</mi></msub><msub><mover><mi>v</mi><mo>~</mo></mover><mi>i</mi></msub><mo>/</mo><mo>(</mo><msub><mi>k</mi><mrow><mtext>cat</mtext><mo>,</mo><mi>i</mi></mrow></msub><msub><mi>f</mi><mi>i</mi></msub><mo>(</mo><mi>x</mi><mo>)</mo><mo>)</mo></mrow></mfrac></mrow><annotation encoding="LaTeX">v_P = \\frac{C}{\\sum_i \\tilde v_i/(k_{\\text{cat},i}f_i(x))}</annotation></semantics></math>.

Note however that this is the production flux per gram biomass. The total biomass may also be constrained by $B\\leq C_2$. The absolute production flux for a single strain is thus

<math xmlns="http://www.w3.org/1998/Math/MathML"><semantics><mrow><msub><mi>J</mi><mi>P</mi></msub><mo>=</mo><msub><mi>C</mi><mn>2</mn></msub><mfrac><mi>C</mi><mrow><msub><mo>∑</mo><mi>i</mi></msub><msub><mover><mi>v</mi><mo>~</mo></mover><mi>i</mi></msub><mo>/</mo><mo>(</mo><msub><mi>k</mi><mrow><mtext>cat</mtext><mo>,</mo><mi>i</mi></mrow></msub><msub><mi>f</mi><mi>i</mi></msub><mo>(</mo><mi>x</mi><mo>)</mo><mo>)</mo></mrow></mfrac></mrow><annotation encoding="LaTeX">J_P= C_2 \\frac{C}{\\sum_i \\tilde v_i/(k_{\\text{cat},i}f_i(x))}</annotation></semantics></math>.

Now, we can divide the reaction pathway over two strains. One could for example take a $v^1$ and $v^2$, such that $v^1+v^2 = v$, where one should note that the metabolites should be transported in between, and this should be written down more carefully than I did here. Again, let $\\tilde v_1$ and $\\tilde v_2$ be normalised such that $\\tilde v_1 + \\tilde v_2$ produces $v_P=1$. Again, we have an enzyme constraint per strain, so we get maximal fluxes per gram biomass of

<math xmlns="http://www.w3.org/1998/Math/MathML"><semantics><mrow><msup><mi>λ</mi><mn>1</mn></msup><mo>=</mo><mfrac><mi>C</mi><mrow><msub><mo>∑</mo><mi>i</mi></msub><msubsup><mover><mi>v</mi><mo>~</mo></mover><mi>i</mi><mn>1</mn></msubsup><mo>/</mo><mo>(</mo><msub><mi>k</mi><mrow><mtext>cat</mtext><mo>,</mo><mi>i</mi></mrow></msub><msub><mi>f</mi><mi>i</mi></msub><mo>(</mo><mi>x</mi><mo>)</mo><mo>)</mo></mrow></mfrac></mrow><annotation encoding="LaTeX">\\lambda^1 = \\frac{C}{\\sum_i \\tilde v^1_i/(k_{\\text{cat},i}f_i(x))}</annotation></semantics></math>,

and

<math xmlns="http://www.w3.org/1998/Math/MathML"><semantics><mrow><msup><mi>λ</mi><mn>2</mn></msup><mo>=</mo><mfrac><mi>C</mi><mrow><msub><mo>∑</mo><mi>i</mi></msub><msubsup><mover><mi>v</mi><mo>~</mo></mover><mi>i</mi><mn>2</mn></msubsup><mo>/</mo><mo>(</mo><msub><mi>k</mi><mrow><mtext>cat</mtext><mo>,</mo><mi>i</mi></mrow></msub><msub><mi>f</mi><mi>i</mi></msub><mo>(</mo><mi>x</mi><mo>)</mo><mo>)</mo></mrow></mfrac></mrow><annotation encoding="LaTeX">\\lambda^2=\\frac{C}{\\sum_i \\tilde v^2_i/(k_{\\text{cat},i}f_i(x))}</annotation></semantics></math>,

where $\\lambda^j$ denotes the number of times that $\\tilde v_1$ can run per gram biomass of strain 1. Since we need to add $\\tilde v^1$ and $\\tilde v^2$ in equal amounts to get the production of $v_P=1$, one needs

$B^1 \\lambda^1 = B^2 \\lambda^2$,

with $B^1 + B^2 = C_2$. This means that $B^1/B^2 = \\lambda^2/\\lambda^1$, and inserting this in the total biomass constraint gives

<math xmlns="http://www.w3.org/1998/Math/MathML"><semantics><mrow><msub><mi>C</mi><mn>2</mn></msub><mo>=</mo><msup><mi>B</mi><mn>1</mn></msup><mo>+</mo><msup><mi>B</mi><mn>2</mn></msup><mo>=</mo><msup><mi>B</mi><mn>2</mn></msup><mfrac><msup><mi>λ</mi><mn>2</mn></msup><msup><mi>λ</mi><mn>1</mn></msup></mfrac><mo>+</mo><msup><mi>B</mi><mn>2</mn></msup><mo>=</mo><msup><mi>B</mi><mn>2</mn></msup><mo>(</mo><mn>1</mn><mo>+</mo><mfrac><msup><mi>λ</mi><mn>2</mn></msup><msup><mi>λ</mi><mn>1</mn></msup></mfrac><mo>)</mo></mrow><annotation encoding="LaTeX">C_2 = B^1 + B^2 = B^2\\frac{\\lambda^2}{\\lambda^1} + B^2 = B^2 (1+ \\frac{\\lambda^2}{\\lambda^1})</annotation></semantics></math>,

so that

<math xmlns="http://www.w3.org/1998/Math/MathML"><semantics><mrow><msup><mi>B</mi><mn>2</mn></msup><mo>=</mo><mfrac><msub><mi>C</mi><mn>2</mn></msub><mrow><mo>(</mo><mn>1</mn><mo>+</mo><mfrac><msup><mi>λ</mi><mn>2</mn></msup><msup><mi>λ</mi><mn>1</mn></msup></mfrac><mo>)</mo></mrow></mfrac></mrow><annotation encoding="LaTeX">B^2 = \\frac{C_2}{(1+ \\frac{\\lambda^2}{\\lambda^1})}</annotation></semantics></math>.

The total production is given by

<math xmlns="http://www.w3.org/1998/Math/MathML"><semantics><mrow><msub><mi>J</mi><mi>P</mi></msub><mo>=</mo><msup><mi>B</mi><mn>1</mn></msup><msup><mi>λ</mi><mn>1</mn></msup><mo>=</mo><msup><mi>B</mi><mn>2</mn></msup><msup><mi>λ</mi><mn>2</mn></msup><mo>=</mo><mfenced><mfrac><msub><mi>C</mi><mn>2</mn></msub><mrow><mo>(</mo><mn>1</mn><mo>+</mo><mfrac><msup><mi>λ</mi><mn>2</mn></msup><mrow><mi>l</mi><mi>a</mi><mi>m</mi><mi>b</mi><mi>d</mi><msup><mi>a</mi><mn>1</mn></msup></mrow></mfrac><mo>)</mo></mrow></mfrac></mfenced><mfenced><msup><mi>λ</mi><mn>2</mn></msup></mfenced><mo>=</mo><mfrac><msub><mi>C</mi><mn>2</mn></msub><mrow><mn>1</mn><mo>/</mo><msup><mi>λ</mi><mn>1</mn></msup><mo>+</mo><mn>1</mn><mo>/</mo><msup><mi>λ</mi><mn>2</mn></msup></mrow></mfrac></mrow><annotation encoding="LaTeX">J_P = B^1 \\lambda^1 = B^2 \\lambda^2 = \\left(\\frac{C_2}{(1+ \\frac{\\lambda^2}{lambda^1})}\\right)\\left(\\lambda^2\\right)=\\frac{C_2}{1/\\lambda^1+1/\\lambda^2}</annotation></semantics></math>,

where we can insert the expression for the maximal possible $\\lambda^1$ and $\\lambda^2$ to get:

$J_P = \\frac{C_2C}{\\sum_i \\tilde v^1_i/(k_{\\text{cat},i}f_i(x)) + \\sum_i \\tilde v^2_i/(k_{\\text{cat},i}f_i(x))} = \\frac{C_2C}{\\sum_i \\tilde v_i/(k_{\\text{cat},i}f_i(x))}$,

which is exactly equal to the production that could be achieved with DoL.

#### Remarks Methods

The Methods-section is clear and covers all prerequisite knowledge without becoming excessively long or boring. The pseudocode is written very clearly, and will definitely help potential users of ASTHERISC. Several strong assumptions are made in the Methods section, and I feel the following should at least be justified a lot more:

##### Equation (3): <math xmlns="http://www.w3.org/1998/Math/MathML"><semantics><mrow><mi>N</mi><mi>r</mi><mo>=</mo><mn>0</mn></mrow><annotation encoding="LaTeX">Nr=0</annotation></semantics></math>

Of course, this seems like a standard assumption that is always made in stoichiometric modelling. However, in this paper the authors are considering the thermodynamic effects that in part depend on metabolite concentrations. The correct formula to use is $Nr=\\mu c$, where $\\mu$ is the growth rate, and $c$ is the vector of metabolite concentrations. This dilution term implies that all metabolites should have some net production when the growth rate is positive. However, is this small net production still thermodynamically feasible in the solutions that ASTHERISC finds? ASTHERISC might just select a really high metabolite concentration, and it will detect no problem because there does not have to be any net production of this metabolite. I wonder if the authors foresee any problems, or if this issue is negligible. This might be important to add in the text. It is true that this issue is resolved by the following assumption that the growth rate is 0, but this might just be the issue: the dilution of metabolites is one of the reasons that I would say it is unreasonable to set the growth rates to 0.

##### The growth rate of all strains is 0

Now this is a rather strong assumption to make, and, as a reader, I really need to be convinced that this is justified. For example, if ASTHERISC assumes that biomass production is zero, how do I know that the algorithm doesn't choose a division of reactions that makes biomass production impossible for one of the strains? For example, it might be thermodynamically ideal to have a high concentration of ATP in one strain, and a low concentration in another. But can the strain with low ATP still grow? Do the biomass-producing reactions not become thermodynamically infeasible? When an organism is unable to grow, is the chosen division of labor still useful for biotechnological applications?

My suggestion is that the authors explicitly state these type of problems. Finding all possible Divisons of Labors is still useful, if they can afterwards be tested for their feasibility. For example, testing if growth is still possible, and if the suggested metabolite exchanges are possible (as the authors do point out in the Dicussion). I would like to see for which multi-strain solutions presented in the paper, growth at these metabolite concentrations is still possible.

##### All strains have identical fractions

Just like in the example in the Introduction, this assumption feels very limiting, but it might not be. Maybe the authors can justify this assumption, but this is not done at the moment. Now, the reader has to think about the consequences of this assumption, and I think that this is actually a task for the authors. My suggestion is that the authors could point out that they focus on the thermodynamic feasibility of a pathway, and not on resource-allocation. Maybe, the multiplication of all reaction rates of one strain is therefore allowed, and then the biomass fraction of the strain does not matter.

To get a feel for the consequences of this assumption, I suggest that the authors write down the full community model without assuming identical fractions (possibly in the SI). Then, the authors may look at the consequences of multiplying the biomass fraction of strain $i$ with a scalar: $F_i \\rightarrow \\lambda_i F_i$. It will turn out that to keep statisfying the steady-state constraint, all reaction rates of strain $i$ must then be divided by $\\lambda_i$. It can then be seen where this will have consequences. I suspect the most important influence in on *satisfying the inhomogeneous flux bounds*. The authors may then explain why this is acceptable.

Probably, the authors can argue that leaving this flexibility of choosing the biomass fractions out only takes away flexibility from the model. So, something that is thermodynamically feasible in this constrained case, will also be feasible in the general case. This seems to be true, as long as the authors do not consider kinetics and limited cellular resources.

#### Remarks Results

The results of applying ASTHERISC to example models are presented. The models are well-chosen to illustrate the workings of the model, and the results are exposed in a clear and un-assuming fashion. The example of the community solution serves its purpose: it illustrates the principle underlying ASTHERISC, and its complexity shows why these results cannot be obtained without a computational toolbox.

##### Page 15

"Reactions for which no $\\Delta_rG'^0$ could be found or calculated (93 reactions (18%) in EColiCore2 and 646 (24%) in iML1515) were blocked in the model (rate fixed to zero)" There are several things I'm wondering about after reading this comment. Why is the chosen solution of blocking these reactions the most reasonable? What are the consequences? Is there any particular property of these reactions which explains why the standard Gibbs free energy cannot be found or calculated?

##### Bottom page 15

"Interestingly, the highest MDF improvements occur when high product yields are demanded. This indicates that low-yield pathways in the single strain might have high MDFs which are more difficult to improve by a community solution." I agree that this is an interesting note.

##### Examplary community solution

The authors state that "kdo8p is of interest as its synthase (KDOPS) has been evaluated as a potential antibiotics target". This does not sufficiently explain to me why kdo8p is of interest. How can the over-production of kdo8p be used to inhibit its synthase? It is not that I do not believe that the authors are correct, I just need a little more explanation.

I don't understand why the concentration of dha is decreased in ecoli2. The authors state that it is used to facilitate the (reverse) PGI, G6PDH2r and PGL reactions, but I don't see this. dha seems to only be used as a substrate in ecoli2, so why would it not be favourable to maximise its concentration, so that g3p-concentration can be decreased even more. Similarly, one would expect dha to be decreased in ecoli1.

#### Remarks Discussion

The discussion is clear and addresses some important points. Especially the paragraph starting with "The application of our algorithm to a metabolic core and a genome-scale model" is very important. Some additional questions that I had with the eventual application of the algorithm:

- I agree that reactions can be knocked-out, but will the optimisation of metabolite concentrations happen automatically? The genetically altered organism has not been evolved to use its new reaction set efficiently, so how do we know that the metabolite concentrations that make the production pathway thermodynamically feasible will be selected?

- Is there a way to add the energy expenditure needed for exchanging metabolites into the algorithm? I guess, maybe naively, that when a metabolite is exported from a low concentration, and imported to a high concentration, at least one of the transport steps needs to use energy. This means that it should at least be thermodynamically feasible for this strain to produce energy.

- Is there a way in ASTHERISC to select the metabolites that may be exchanged by the algorithm? This does not seem hard to implement.

In addition, I would like a discussion of the kinetic analog of this thermodynamic problem. Except for making pathways thermodynamically feasible, Divison of Labor could also help with the efficient usage of cellular resources. Indeed, as the authors show in the Introduction, not when the metabolite concentrations are unchanged, but what if these concentrations are changed. Can this problem also be addressed? If not, why not? Or is it just less important according to the authors?

#### Minor comments

**Figure 1** I think Figure 1 is rather messy. A reader that did not understand the explanation in the text, will probably not be aided by this figure a lot. This is a matter of taste though, not of right and wrong.

**Caption Figure 2** Add comma: "would follow, indicating"

**After Equation 5** Sentence starting with "Furthermore, the biomass fractions" is pretty hard to read.

**Equation 9** It is not immediately clear whether the $i$-th column is taken before or after the transpose

**Equation 10** I fail to see why the notation is different between Equations (9) and (10): $\\Delta_r G_i' \\rightarrow \\Delta G_i$ and $\\hat N \\rightarrow N$. These might be typos. If not, more explanation might be needed.

**Page 11: "A specified minimal fraction of this maximum yield must be reached in all subsequent community model calculations"** Can this fraction also be chosen larger than 1? Or is the maximum possible yield determined without taking into account the thermodynamic feasibility? If so, please indicate this in the text.

**Page 14** Typo in: "had to be separated in to strains"

**Page 15** Typo: "For each of the producible target metabolite"

**Page 17** "Some of these dominant MDF values occurred in all analyzed models." So, there are MDF values that appear often in the core- and in the genome-scale model? If so, it might be interesting to explain a bit more why this is.

**The use of "exemplary"** It might be that I'm wrong here, but it seems that the word 'exemplary' is used meaning that something is an example, but my interpretation of this word is that it is "a perfect example". It therefore seems a bit like bragging to me when this word is used.

**Exemplary community solution** "In the community solution found by ASTHERISC, the MDF value can be increased by 26 % up to 2.87 kJ/mol reaching a yield of 0.733 mol/mol" It confuses me that this yield is here repeated, it seems like this yield is different than the yield of the single-strain model. But this was exactly the yield that was requested, right? So, the reported MDF value for the single-strain model also attains this yield?

Reviewer #3: Review on „Designing microbial communities to maximize the thermodynamic driving force for the production of chemicals by Bekiaris and Klamt

Currently, investigations of mixed synthetic cultures is in the focus of experimental and theoretical studies in biotechnology. Mixed cultures have certain advantages over single species cultivation (for example division of labour) and therefore, the potential of this approach becomes important for the design of processes in various applications. Synthetic mixed cultures could be seen as a mixture of two different species or a mixture of the same species with defined mutations. The manuscript is based on the second variant and analysis the steady-state behaviour of two or three different cultures form a thermodynamic point of view; while standard tools for analysis of metabolic networks like FBA (and its extensions) usually determine intracellular fluxes, more sophisticated tools take into account thermodynamic limitations only for single species. Bekiaris and Klamt ask on a meaningful division of the metabolic network such, that the production of selected compounds of interest is improved (or even first possible) by using more than one strain variant. The criterion used is the MDF, that is, the Max-min-driving force (the maximization of the minimal negative value of the Gibbs energy of each reaction of the entire network/ pathway). The study presented by Bekiaris and Klamt is accurate and comprehensive, and I only have a minor number of comments to improve the manuscript.

(i) How can the minimal OptMDF advantage (0,2 kJ/mol) be interpreted? Are there any reasons for taking just this value?

(ii) Are there any constraints on the uptake rate of glucose and other main components from the environment?

(iii) The authors have chosen models with a higher number of reactions and components. For many applications core models are applied? How would the results look like if a standard core model (for example from the COBRA toolbox or the compressed version of the Ecolicor2 with only 82 reactions and 54 metabolites) is used? This is of special interest in light of the result that most exchange metabolites are found in glycolysis or in pathways closely related to glycolysis.

(iv) A basic assumption is that the growth rate for all species is the same. I miss a discussion of the influence of the growth rate on the results. What is the value of the growth rate for the example presented in detail (kdo8p)? Strain ecoli1 takes up glucose and I expect a high growth rate, and I wonder why - in practice - a substrate like dha should be taken up by ecoli2 even in the case a transport system is available?

Is it implemented that the growth rate must be equal for both strains (maybe I missed this constrain in the Method section)?

___

Minor comment: while I could reproduce the value of Figure 2 right, I found a different value for Figure 2 left (-0,245 instead of -0,19).

**Have all data underlying the figures and results presented in the manuscript been provided?**

Reviewer #1: Yes

Reviewer #2: Yes

Reviewer #3: Yes

PLOS authors have the option to publish the peer review history of their article (what does this mean?). If published, this will include your full peer review and any attached files.

Reviewer #1: **Yes: **Daniel Machado

Reviewer #2: **Yes: **Daan de Groot

Reviewer #3: No
---

## [Decision Letter · Decision Letter 1]

18 May 2021

Dear Dr. Klamt,

We are pleased to inform you that your manuscript 'Designing Microbial Communities to Maximize the Thermodynamic Driving Force for the Production of Chemicals' has been provisionally accepted for publication in PLOS Computational Biology.

Before your manuscript can be formally accepted you will need to complete some formatting changes, which you will receive in a follow up email. A member of our team will be in touch with a set of requests. Moreover, there is one final comment by reviewer 3 which I expect you to address along with these formatting changes.

Best regards,

Christoph Kaleta

Associate Editor

PLOS Computational Biology

Daniel Beard

Deputy Editor

PLOS Computational Biology

Reviewer's Responses to Questions

**Comments to the Authors:**

Reviewer #1: The authors have properly addressed all my comments.

Reviewer #2: The reviewers addressed all my raised concerns, so I advise publication.

Reviewer #3: Second review

The authors responded satisfactorily to my questions. However, based also on the comments of other reviewers, I recommend to make a clear statement already in the abstract and in the introduction that a scenario is considered where the growth rate is set to zero. This is only mentioned incidental but has a tremendous effect on the outcome of the simulation results. Since a basic requirement is that the growth rates are equal, a number of additional constraints appear and I would like to see a deeper discussion on this issue in the manuscript. Currently, the already added paragraph in the discussion section is not enough.

**Have the authors made all data and (if applicable) computational code underlying the findings in their manuscript fully available?**

Reviewer #1: None

Reviewer #2: Yes

Reviewer #3: Yes

PLOS authors have the option to publish the peer review history of their article (what does this mean?). If published, this will include your full peer review and any attached files.

Reviewer #1: **Yes: **Daniel Machado

Reviewer #2: **Yes: **Daan de Groot

Reviewer #3: No

---

## [Editor Report · Acceptance letter]

9 Jun 2021

PCOMPBIOL-D-21-00273R1 

Designing Microbial Communities to Maximize the Thermodynamic Driving Force for the Production of Chemicals

Dear Dr Klamt,

I am pleased to inform you that your manuscript has been formally accepted for publication in PLOS Computational Biology. Your manuscript is now with our production department and you will be notified of the publication date in due course.

With kind regards,

Katalin Szabo
